# Identification and Characterization of α-Glucosidase Inhibition Flavonol Glycosides from Jack Bean (*Canavalia ensiformis* (L.) DC

**DOI:** 10.3390/molecules25112481

**Published:** 2020-05-27

**Authors:** Anita M. Sutedja, Emiko Yanase, Irmanida Batubara, Dedi Fardiaz, Hanifah N. Lioe

**Affiliations:** 1Faculty of Applied Biological Sciences, Graduate School of Applied Biological Sciences, Gifu University, 1-1 Yanagido, Gifu City, Gifu 501-1193, Japan; maya@ukwms.ac.id; 2Department of Food Science and Technology, Faculty of Agricultural Engineering and Technology, IPB University, Kampus IPB Dramaga, Bogor 16680, West Java, Indonesia; dedi_fardiaz@yahoo.com.sg (D.F.); hanifahlioe@apps.ipb.ac.id (H.N.L.); 3Department of Food Technology, Faculty of Agricultural Technology, Widya Mandala Catholic University Surabaya, Jalan Dinoyo 42-44, Surabaya 60265, East Java, Indonesia; 4Department of Chemistry, Faculty of Mathematics and Natural Science, IPB University, Kampus IPB Dramaga, Bogor 16680, West Java, Indonesia; imebatubara@gmail.com; 5Tropical Biopharmaca Research Center IPB University, Kampus IPB Taman Kencana No 3, Bogor 16128, West Java, Indonesia

**Keywords:** *Canavalia ensiformis*, legume, jack bean, constituent profiling, flavonol glycosides, flavonoid

## Abstract

Although the intake of jack bean (*Canavalia ensiformis* (L.) DC.), an underutilized tropical legume, can potentially decrease the risk of several chronic diseases, not much effort has been directed at profiling the polyphenolics contained therein. Hence, this work aimed to identify and quantify the dominant jack bean polyphenolics, which are believed to have antioxidant and other bioactivities. Four major compounds were detected and identified as kaempferol glycosides with three or four glycoside units. Their structures were established based on UV-visible, 1D, 2D NMR, and HR-ESI-MS analyses. Specifically, kaempferol 3-O-α-l-rhamnopyranosyl (1→6)- β-d-glucopyranosyl (1→2)-β-d-galactopyranosyl-7-O-[3-O-*o*-anisoyl]-α-l-rhamnopyranoside was detected for the first time, while the other three compounds have already been described in plants other than jack bean. This new compound was found to have a higher α-glucosidase inhibition activity compared to acarbose.

## 1. Introduction

Diabetes is one of the diseases whose global patient prevalence has increased every year. Persons with diabetes are predicted to increase from 463 million in 2019 to 700 million in 2045, a 51% increase, mainly from increases in low and middle income countries [1]. Type 2 diabetes accounts for the vast majority (around 90%) of diabetes worldwide. This disease refers to the insufficient of insulin uptake of glucose in blood, causing a high blood glucose level. It can lead to complications in many parts of the body and develop some serious life-threatening health problems when not well managed. 

Decreasing postprandial hyperglycemia has become one of the effective ways to manage diabetes mellitus, in particular, non–insulin-dependent diabetes mellitus (NIDDM). During carbohydrates metabolism, α-glucosidase is the key enzyme catalyzing the final step in the digestive process as liberated d-glucose from dietary complex carbohydrates [2]. Inhibition of carbohydrate hydrolyzing enzymes, such as α-glucosidase and α-amylase, in the digestive organs retard the absorption of glucose. As the result, there are reduced postprandial plasma glucose levels and suppression of postprandial hyperglycemia [3]. However, using drugs for diabetes management may have several negative side effects for the patients, such as cardiovascular disease and morbidity [4,5]. 

In recent years, many efforts have been made to identify effective α-glucosidase inhibitors from natural sources in order to transiently lower the blood glucose, preventing heart disease and high blood pressure, also enhancing the antioxidant system, insulin action, and secretion [6]. this compound might be applied in development of a functional food or lead compounds for use against diabetes. Polyphenols, mainly flavonoids and phenolic acids [7], have been isolated from plants found as α-glucosidase inhibitors. Parts of those plants are edible and are processed and consumed daily, such as fruits, vegetables, cereals, and legumes.

Legumes such as soybeans, peanuts, mung beans, and red kidney beans are major sources of dietary protein for millions of people in developing countries, including Indonesia. Jack bean (*Canavalia ensiformis* (L.) DC.) is one of the legume plants cultivated in Indonesia and other parts of East Asia. The high protein content of this bean makes them a promising food [8,9]. In addition, this bean is easy to cultivate due to its high adaptability to adverse conditions. On the other hand, jack bean is actually underutilized as food, although commonly used as a fertilizer and animal feed. This is because jack bean seeds contain toxic and antinutritional compounds such as concanavalin A and are therefore of limited use as human food or animal feed [10,11,12,13]. However, the content of these harmful compounds can be easily reduced by heating [10,14,15,16] and non-heating processes such as germination and using chemicals [15].

In addition, jack bean seeds contain saponins, flavonoids, and alkaloids [17], as well as certain polyphenols that exhibit antioxidant activity and can potentially decrease the risk of several chronic diseases [8,18,19], include diabetes mellitus. Nevertheless, not much is currently known about the type and amounts of polyphenols in jack bean. In view of the above, the present work aims to identify, quantify the dominant polyphenolics in jack bean, and investigate its potency as an antidiabetic agent.

## 2. Results and Discussion

Jack bean flour was extracted with MeOH, and the extract was subjected to reverse-phase column chromatography using sequential elution with 20, 50, 80, and 100 vol% aqueous MeOH to obtain four fractions (F20, F50, F80, and F100, respectively). HPLC analysis of these fractions revealed the presence of multiple peaks in F50 and F80, which were therefore subjected to further separation (Figure 1). F50 was subjected to normal-phase column chromatography to afford **1** and **2** as major products, while F80 was subjected to reverse-phase preparative chromatography to afford **3** and **4**. The chemical structures of the isolated compounds were investigated and identified through intensive spectroscopic analyses based on 1D and 2D nuclear magnetic resonance (NMR), high-resolution electrospray ionization mass spectrometry (HR-ESI-MS), and gas chromatography mass spectrometry (GC-MS) data as follows (Appendix A).

### 2.1. Structure Elucidation of ***1***–***4***


According to HR-ESI-MS (*m/z* 887.2843, [M + H]^+^), **1** had a molecular formula of C_39_H_50_O_23_ (calcd. for [C_39_H_51_O_23_]^+^, 887.2821). The corresponding ^1^H NMR spectrum (Table 1) of this compound assigned a flavanol based on the exhibited signals at 6.81 and 8.00 ppm corresponding to the four aromatic protons (AA’BB’) of B ring, and signals at 6.36 and 6.63 ppm corresponding to the two meta-aromatic protons of A ring. ^13^C NMR (Table 1) and heteronuclear multiple quantum coherence spectroscopy (HMQC) data suggested the presence of 39 carbons, including one carbonyl carbon (179.52 ppm), eight non-protonated carbons, twenty-six methine carbons, one methylene carbon, and three methyl carbons. Heteronuclear multiple bond correlation spectroscopy (HMBC) analysis revealed a long-range correlation between the ^1^H signal at 8.00 ppm (H-2’ and 6’) and the ^13^C peak at 159.22 ppm (C2), suggesting the presence of a kaempferol skeleton. 

Additionally, ^1^H NMR signals at 3.21–3.93 ppm indicated the presence of sugars. Four anomeric proton signals at 5.53, 5.12, 4.42, and 5.47 ppm corresponded to four sugar units, three of which were α-rhamnosyl moieties, as exemplified by the signals of the methyl group in position six at 0.88, 1.07, and 1.16 ppm. The other sugar unit was hexose and indicated as glucose because of a large coupling constant (6.4 Hz) detected at position C5 (3.54 ppm) of its structure. 

Two rhamnose moieties were assumed to be bonded to a glucose moiety on the basis of HMBC data. The anomeric proton of rhamnose (5.12 ppm) bound to the second carbon (77.57 ppm) of the glucose structure, while the 4.42 ppm anomeric proton from rhamnose bound to the sixth carbon (67.15 ppm) of the same glucose structure. HMBC cross peaks were observed between the anomeric proton of glucose and the C3 (134.72 ppm) and of rhamnose and C7 (163.40 ppm) of the kaempferol moiety. 

To confirm the suggested sugar subunit structure, **1** was subjected to acid hydrolysis, and the hydrolysate was analyzed by GC-MS to reveal the presence of α-l-rhamnose and β-d-glucose (Appendix A). Based on the combined data, **1** was identified as kaempferol 3-O-α-l-rhamnopyranosyl (1→2) [α-l-rhamnopyranosyl (1→6)]-β-d-glucopyranoside-7-O-α-l-rhamnopyranoside (Figure 2). This compound was previously isolated from *Styphnolobium japonicum* leaves [20] but has not been reported to be present in jack bean. All data agreed with those published for this compound previously.

Based on HR-ESI-MS data (*m/z* 741.2214, [M + H]^+^), **2** was assigned the molecular formula of C_33_H_40_O_19_ (calcd. for [C_33_H_41_O_19_]^+^, 741.2242), and the corresponding ^1^H and ^13^C NMR data suggested the presence of a kaempferol skeleton as the aglycone. The signals of three anomeric protons at 4.52, 5.21, and 5.60 ppm suggested the presence of three sugar units, while resonances at 0.97 and 1.17 ppm were assigned to methyl in position six of rhamnose moieties. Taking into consideration the values of coupling constants, the sugar units were identified as α-rhamnose and β-glucose. HMBC cross peaks were observed between the anomeric proton of glucose and C-3 position of the kaempferol moiety, and GC-MS analysis confirmed the presence of L-rhamnose and D-glucose units (Appendix A). Thus, **2** was identified as kaempferol 3-O-(2,6-di-α-L-rhamnopyranosyl)- β-D-glucopyranoside (Figure 2), which has already been isolated from petals of *Clitoria ternatea* [21].

Based on HR ESI-MS data (*m/z* 1037.3076 [M + H]^+^), **3** was assigned the formula of C_47_H_56_O_26_ (calcd. for [C_47_H_57_O_26_]^+^, 1037.3138) and was concluded to have the same aglycone (kaempferol) as **1** and **2**. ^1^H NMR and homonuclear correlation spectroscopy (COSY) data suggested the presence of an aromatic moiety other than that of kaempferol, which was identified as anisoyl on the basis of ^13^C NMR, HMQC, and HMBC data. Four sugar moieties were identified based on the signals of four anomeric protons at 5.63, 4.77, 4.51, and 5.63 ppm. Two methyl carbons and ^1^H signals at 1.17 and 1.33 ppm were indicative of the existence of rhamnose, while the signals of two methylene carbons referred to position six of each hexose structure. The other two sugar units were presumed to be galactose and glucose by the coupling constant at C5 position. HMBC cross peaks between the carbonyl carbon (167.38 ppm) and the proton signal at 5.34 ppm showed the presence of an anisoyl group bonded to the C3 position of rhamnose. Moreover, HMBC cross peaks also allowed one to deduce the connection sequence of two hexose moieties and rhamnose (terminal position). Anomeric proton of rhamnose (4.51 ppm) was connected to C6 (62.56 ppm) in the glucose structure, while its anomeric proton (4.77 ppm) was connected to C2 (80.14 ppm) in the galactose structure so that those three sugars were connected sequentially. Galactose was bound to the C3 position (135.15 ppm) of kaempferol through its anomeric proton (5.28 ppm).

GC-MS analysis confirmed the presence of L-rhamnose, D-galactose, and D-glucose (Appendix A). Thus, **3** was identified as kaempferol 3-O-α-l-rhamnopyranosyl (1→6)-β-d-glucopyranosyl (1→2)-β-d-galactopyranosyl-7-O-[3-O-*o*-anisoyl]-α-l-rhamnopyranoside (Figure 2). Compound **3** was found to be similar to gladiatoside A_2,_ which was previously isolated from sword bean (*Canavalia gladiata*). Those two compounds had the same mass number and contained the same type of four glycoside units and one anisoyl unit [22]. 

The fundamental differences between these two compounds were in the bonding position between L-rhamnose, D-glucose, and D-galactose that are attached in the C3 position of the kaempferol. In the gladiatoside A_2_, galactose was connected to glucose through a (1→2) linkage and to rhamnose by a (1→6) linkage. Meanwhile in **3**, L-rhamnose was connected to glucose by a (1→6) linkage and D-glucose connected by a (1→2) linkage to D-galactose unit and formed a linear structure of trisaccharide. This sequence connection of galactose, glucose, and rhamnose made **3** different to gladiatoside A_2_ and found it as a novel compound. 

Based on HR-ESI-MS data (*m/z* 1021.3170 [M + H]^+^), **4** was assigned the formula of C_47_H_56_O_26_ (calcd. for [C_47_H_57_O_25_]^+^, 1021.3189), featuring kaempferol as the aglycone, four glycoside moieties, and one anisoyl unit, similarly to **3**. The four sugar units were identified based on the four anomeric proton signals at 5.63, 5.25, 4.51, and 5.62. Proton signals at 1.17, 0.99, and 1.33 ppm corresponded to the 6-position of α-rhamnose moieties. GC-MS analysis confirmed the presence of L-rhamnose and *D*-galactose units (Appendix A). Thus, **4** was identified as kaempferol 3-O-(2,6-di-α-l-rhamnopyranosyl)-β-d-galactopyranoside-7-O-[3-O-*o*-anisoyl]-α-l-rhamnopyranoside (Figure 2), which has previously been isolated as gladiatoside B_2_ from sword bean (*Canavalia gladiata*) [22].

### 2.2. UPLC-ESI-MS/MS Analysis 

The fragmentation patterns (in particular, the neutral loss) of each pseudo molecular ion were used to identify the type and number of sugar residues (Appendix A). Fragments of **1** (Table 2, Appendix A) with *m/z* 741.2214 and 595.1542 corresponded to neutral losses of 146 and 292 Da from the pseudomolecular ion with *m/z* 887.28434 ([M + H]^+^), i.e., reflected the loss of two rhamnosyl residues. In addition, the fragment at *m/z* 433.0958 corresponded to a neutral loss of 454 Da, i.e., to the loss of two rhamnose and glucose units. This data suggested that two rhamnose units bonded to the glucose unit attached to the C3 position. It was because fragmentation occurs most easily in the C3 position in the positive mode. [23,24]. The fact that the relative abundance of ^3^Y_1_^+^ (*m/z* 595.1542) exceeded that of ^3^Y_2_^+^ (*m/z* 741.2214) suggested that one rhamnose was (1→2) linked to the glucose, while the other rhamnose was connected in a (1→6) linkage [23,24,25]. 

Fragmentation in the negative mode explained the presence of one rhamnose moiety attached to C7 position of kaempferol skeleton as showed by the neutral loss of 146 Da between pseudomolecular ion with *m/z* 885.2556 [M − H]^−^ and fragment ion of *m/z* 739.6731. It was reported that the glycoside in the C7 position is easy to be fragmented in negative mode [26,27]. Therefore, **1** contained one glucose unit and three rhamnose units, two of which were bound to the glucose unit. The above data agreed with the structure put forward based on NMR data (Figure 2 and Figure 3). 

Similar fragmentation patterns were observed for **2**–**4**. In addition, fragment ions with *m/z* 135 and 281 for **3** and **4** supported the presence of a rhamnose unit with an anisoyl group (Appendix A). Thus, MS/MS data (Table 2) supported the structure proposed based on NMR data.

The major constituents of F50 and F80, isolated by column chromatography and preparative HPLC, were identified as **1** and **2** (F50) and **3** and **4** (F80). In addition to the four dominant compounds, F50 and F80 contained 10 other compounds at levels that did not allow the isolation of sufficient quantities for NMR analysis (1D and 2D). Compounds **5**–**10**, identified in F50, and compounds **11**–**14**, obtained from F80, exhibited similar UV-vis spectral patterns with compounds **1**–**4** and were therefore presumed to be kaempferol derivatives. Therefore, further identification was carried out using MS/MS analysis. The fragment ion at *m/z* 287 indicated that **5**–**14** contained kaempferol as the aglycone (Table 3, Appendix A).

The glycoside compositions of **5**–**14** were identified as described previously, with MS/MS data for each compound presented in Appendix A and the relative mass number (indicated as [M + H]^+^), aglycone, and glycan composition of each compound listed in Table 3. Compounds **11** and **13** were isomers of **3**, while **12** and **14** were isomers of **4** and were identified as gladiatosides B_1_ and B_3_, respectively [22].

### 2.3. Quantitation of Kaempferol Content in Methanolic Jack Bean Extract 

The kaempferol glycosides content in jack bean extract was determined from HPLC peak areas (Table 3). The overall kaempferol glycoside content (calculated as the total of fourteen compounds) in jack bean flour was 5.57 mg/100 g and compounds **1–4** composed 2.29 mg/100 g (41.05%), 0.67 mg/100 g (11.94%), 0.49 mg/100 g (8.81%), and 0.39 mg/100 g (7.03%). This data explained that this bean was a good source of **1** and in accordance with the highest isolation amount obtained in this research. Furthermore, 52.75% of total kaempferol was kaempferol tetraglycosides, while kaempferol triglycosides was 17.71%. 

### 2.4. In Vitro α-Glucosidase Inhibition Activity 

Inhibitory activity of α-glucosidase is one of the useful parameters to show the potential of compounds **1–4** for preventing diabetes. This enzyme plays a main role in hydrolyzing sugar compounds into monomers that can be easily absorbed in the small intestine. Inhibition activity of the α-glucosidase enzyme by the compounds **1–4** showed in Table 4.

Moderate inhibitory activity of glucosidase was observed in the subfractions F50 and F80. Compounds **1**–**4**, obtained by further purification, exhibited higher inhibition activity and were significantly different (*p* < 0.05) than before separation. This result suggested that compounds **1**–**4** were one of the important active components of each sub fraction. F80 had higher inhibition activity than F50 (Table 4). Compounds **1** and **2**, obtained from F50, had low inhibitory activity compared to acarbose. It has been reported that glycosylation of flavonoids lowered the inhibitory activity through α-glucosidase [28], and our result is in agreement with it. The bigger number of glycosides attached caused a lowering of the inhibition effect [29]. 

Inhibition activity of compound **3** and **4**, obtained from F80. were significantly higher than acarbose, **1** and **2**. Although **3** and **4** were kaempferol glycosides, the presence of anisoyl groups attached to rhamnose moiety at position C7 distinguished these two compounds with **1** and **2**. The presence of anisoyl groups made stronger the inhibition activity of α-glucosidase. As anisoyl group was thought to influence the high activity of α-glucosidase inhibition activity (Table 4), all kaempferol glycosides with anisoyl group may have similar ability to inhibit α-glucosidase activity. As described in Section 2.3, those compounds composed 28.60% (1.59 mg/100 g) of kaempferol glycosides content in jack bean. That described that this bean has potency as an ingredient for functional food. Compound **3** had higher inhibitory activity than **4**, although the number of sugar moiety was same. This result suggested that the difference of binding pattern of sugar moiety might be influenced by the inhibition activity. 

## 3. Materials and Methods 

### 3.1. Plant Materials

Jack beans (*C. ensiformis* (L.) DC.) were purchased from a local farmer located in Temanggung, Central Java. Broken, molded, sprouted, and damaged beans were removed before shipping to the laboratory. This bean had been identified by the Research Center for Biology, Indonesian Institute of Science No. 1761/IPH.1.01/If.07/VII/2017 and stored at Tropical Biofarmaca Research Center with a voucher specimen No. BMK0461012020. The beans were further peeled, ground, and sieved through 100-mesh sieves to prepare jack bean flour that was stored at −30 °C prior to analysis.

### 3.2. Reagents and Standards

For extraction and isolation, n-hexane and methanol were purchased from Kanto Chemicals (Osaka, Japan) and Kieselgel 60 F_254_ purchased from Merck (Darmstadt, Germany) used for thin-layer chromatography (TLC). Deuteromethanol (CD_3_OD) (Kanto Chemical, Osaka, Japan) was used for preparing NMR samples. The mobile phase used for HPLC and UPLC were acetonitrile (Kanto Chemical, Osaka, Japan) and formic acid (FUJIFILM Wako Pure Chemical, Japan). L-cysteine methyl ester hydrochloride and *N*, *O*-bis (trimethylsilyl) trifluoroacetamide (Tokyo Chemical Industry, Tokyo, Japan) and pyridine (Nacalai Tesque, Japan) were used for acid hydrolysis of GC-MS samples. Methanol purchased from Nacalai Tesque used for polarimeter, UV and quantification analysis. Kaempferol as standard for quantification was purchased from Tokyo Chemical Industry (Japan) For IR analysis, KBr was purchased from FUJIFILM Wako Pure Chemical. For the α-glucosidase inhibitory activity assay, α-glucosidase from Saccharomyces cerevisiae was purchased from Sigma Aldrich (Germany). Acarbose hydrate, as the positive control, was obtained from Tokyo Chemical Industry (Tokyo, Japan). Phosphate buffer was prepared by mixing dipotassium hydrogen phosphate and potassium dihydrogen phosphate purchased from FUJIFILM Wako Pure Chemical. Dimethyl sulfoxide, sodium chloride, and sodium carbonate were purchased from Nacalai Tesque.

### 3.3. Extraction and Isolation

Jack bean flour (1000 g) was extracted with 5 L hexane (25 °C, 24 h) using a shaker to remove fats, and the solid residue was isolated by filtration and dried to obtain fat-free bean flour. This flour was then extracted with methanol (3 × 5 L, 25 °C, 2 h), and the extract was isolated by filtration and concentrated under reduced pressure. The residue was suspended in H_2_O (200 mL) and chromatographed on a Daion HP-20SS (Mitsubishi Chemical Corporation, Tokyo, Japan) column. Elution was started with H_2_O to separate sugars, starch, and proteins, and was continued using progressively increasing levels of MeOH in water (20, 50, 80, and 100 vol%, 2 L for each fraction) to afford four fractions (F20, F50, F80, and F100, respectively) that were subjected to HPLC for further identification.

F50 was subjected to silica gel open-column chromatography using chloroform–MeOH–H_2_O (14:6:1 and 13:7:1, each 300 mL) as an eluent to give five sub-fractions, namely F50A–F50E. F50A and F50B were obtained as pure compounds and they were compound **1** (7.0 mg) and **2** (2.7 mg). TLC was performed to monitor the collection of fractions and spots were visualized by dipping in 1% vanillin solution followed by 10-min heating at 110 °C.

F80 was subjected to preparative HPLC using a Cosmosil 5C18-MS-II column (10 mm I.D. × 250 mm, particle size of 5 μm) maintained at 35 °C. Preparative HPLC was performed using a Jasco PU2089 intelligent pump equipped with a Jasco UV-2075 detector (Tokyo, Japan) and a Shimadzu CTO-10ACVP column oven (Kyoto, Japan). Elution was performed with 20% MeCN and 0.5% HCOOH in water at a flow rate of 5.0 mL/min to obtain **3** (4.4 mg) and **4** (2.8.mg). Chromatogram was monitored by UV absorption at wavelength of 254 nm. The chemical structure of compounds **1**–**4** were determined by combination of NMR and MS/MS analysis. Determination of compounds **5**–**14** were performed by MS/MS analysis only.

### 3.4. HPLC Condition

F50 and F80 samples were dissolved in MeOH and injected (5 μL) into a reverse-phase HPLC-PDA system consisted of Jasco PU2089 intelligent pump equipped with a JASCO MD-2010 plus detector and a Jasco CO-2065 Plus column oven (Tokyo, Japan). Reversed phase column (Cosmosil 5C18-MS-II, 4.6 ID × 150 mm, particle size of 5 μm) was used and the temperature was controlled at 35 °C. F50 was performed using mobile phase contained 0.5% HCOOH in H_2_O (A) and acetonitrile (B), and the following gradient was used: 10% B for 10 min, 15% B at 30 min, 25% B at 45 min, 35% B at 55 min, 45% B at 60 min, and 55% B at 70 min. F80 was performed using 20% MeCN and 0.5% HCOOH in water on isocratic condition. The flow rate was set to 1 mL/min, and detection was performed at 254 nm. Chromatograms of F50 and F80 as shown on Figure 1a,b.

### 3.5. Acid Hydrolysis of ***1***–***4***

Hydrolysis was performed as described elsewhere [22]. Compounds **1**–**4** were dissolved in 5% aq. H_2_SO_4_–1,4-dioxane (1:1, *v*/*v*; 1 mL), and the solution was stirred under reflux for 2 h, cooled, and neutralized using Amberlite IRA-400 (OH-form) (Sigma Aldrich, St. Louis, MO, USA), which was then removed by filtration. The filtrate was concentrated, and the residue was loaded on a Monospin C18 spin (GL science, Japan) column and eluted with H_2_O and MeOH. The H_2_O eluate was concentrated, and the residue was sequentially treated with L-cysteine methyl ester hydrochloride (3 mg) in pyridine (0.5 mL) at 60 °C for 1 h and *N*, *O*-bis (trimethylsilyl)trifluoroacetamide (0.2 mL) at 60 °C for 1 h. The supernatant was further analyzed by GC-MS.

### 3.6. GC-MS Condition

An Agilent 6890N gas chromatograph (Agilent Technologies, Santa Clara, CA, USA) fitted with a DB-5 column (30m, 250 µm I.D., film thickness 0.25 µm, J&W Scientific, PA, USA) coupled to a JMS-AMSUN200/GI UltraQuad GC-MS instrument (JEOL Ltd., Tokyo, Japan) was employed. The column temperature was set to 280 °C, and the He flow rate equaled 1 mL/min. The oven temperature was increased from 100 to 180 °C at 10 °C/min, then to 240 °C at 25 °C/min, held for 10 min, increased to 300 °C at 25 °C/min and held for 2 min. The MS scan range equaled 50–1000, and scanning was performed at a rate of 2 scans/s. The transfer line and ion source were held at 280 °C. The MS spectrum of each peak was identified using the library match software from the National Institute of Standards and Technology (NIST) library and compared to authentic standards.

### 3.7. Ultrahigh-Performance Liquid Chromatography–Time-of-Flight Mass Spectrometry (UPLC-TOF-MS) Conditions

Analysis was performed on an UPLC system coupled to a QTOF-MS (Waters Xevo G2 QTof, Waters, Milford, MA, USA) instrument operated in electrospray ionization (ESI) mode at a mass resolution of 20,000 and controlled by MassLynx 4.1 software. An Acquity UPLC BEH C18 column (2.1 mm I.D. × 100 mm, 1.7 μm, Waters, USA) at 35 °C was used for chromatographic separation. The sample (1 μL) was injected using an autosampler. The mass spectrometer was calibrated with 0.5 mM sodium formate. Leucine enkephalin (2 µg/mL, *m/z* 556.2771 in positive mode) was used as lock spray at a flow rate of 10 µL/min. The collision energy equaled 6 V. The source parameters were as follows: capillary voltage 2.5 kV, sampling cone voltage 30 V, extraction cone voltage 4 V, source temperature 150 °C, desolvation temperature 500 °C, desolvation gas flow 1000 L/h, cone gas flow 50 L/h. Each compound was fragmented using a range of 25–30 V for MS/MS scans. The mobile phase contained 1% HCOOH in H_2_O (A) and MeCN (B). F50 was separated using the following gradient: initial 10% B for 1.2 min, 15% B at 3.5 min, 25% B at 5.2 min, 35% B at 6.4 min, 45% B at 7.0 min, 55% B at 8.1 min, hold until 10 min. F80 was separated isocratically using 20% B for 50 min.

### 3.8. Spectroscopic Data of Compounds ***1***–***4***

#### 3.8.1. Kaempferol 3-O-α-l-rhamnopyranosyl (1→2) [α-l-rhamnopyranosyl (1→6)]-β-d-glucopyranoside-7-O- α-l-rhamnopyranoside (**1**)

Yellow powder. ${\left[\alpha \right]}_{D}^{23}$ = −33.15° (c = 0.70, MeOH) was performed on a Jasco P2300 digital polarimeter (Jasco Corporation, Tokyo, Japan). HR-ESI-MS: *m*/*z* 887.2843 [M + H]^+^ (calcd. for [C_39_H_51_O_23_]^+^, 887.2821) was carried out using a Waters Xevo G2 QTOF LC/MS system (Waters, Milford, MA, USA). UV λmax/nm (log ε): 267 (4.71), 350 (4.60) were recorded on a Lambda 950 UV-vis-NIR spectrometer (Perkin Elmer, Shelton, CT, USA) in MeOH solution. IR νmax/cm^−1^: 1655, 3402 was recorded on a Spectrum 100 FT-IR spectrometer (Perkin Elmer, Shelton, CT, USA) using KBr as a matrix. 1H and 13C NMR spectra were recorded in CD_3_OD on Bruker AV800 (800 MHz) instruments (Bruker, MA, USA): see Table 1.

#### 3.8.2. Kaempferol 3-O-(2,6-di-α-l-rhamnopyranosyl)-β-d-glucopyranoside (**2**)

Yellow powder. ${\left[\alpha \right]}_{D}^{23}$ = −32.15° (*c* = 0.27, MeOH). HR ESI-MS: *m/z* 741.2214 [M + H]^+^ (calcd. for [C_33_H_41_O_19_]^+^, 741.2242). UV *λ*_max_/nm (log *ε*): 268 (4.87), 348 (4.63). IR *ν*_max_/cm^−1^: 1657, 3402. ^1^H and ^13^C NMR (CD_3_OD): see Table 1.

#### 3.8.3. Kaempferol 3-O-α-l-rhamnopyranosyl (1→6)-β-d-glucopyranosyl (1→2)-β-d-galactopyranosyl-7-O-[3-O-o-anisoyl]-α-l-rhamnopyranoside (**3**)

Yellow powder. ${\left[\alpha \right]}_{D}^{23}$ = −64.77° (*c* = 0.44, MeOH). HR ESI-MS: *m/z* 1037.3076 [M + H]^+^ (calcd. for [C_47_H_57_O_26_]^+^, 1037.3138). UV *λ*_max_/nm (log *ε*): 268 (5.10), 349 (4.96). IR *ν*_max_/cm^−1^: 1602, 3436. ^1^H and ^13^C NMR (CD_3_OD): see Table 1.

#### 3.8.4. Kaempferol 3-O-(2,6-di-α-l-rhamnopyranosyl)-β-d-galactopyranoside-7-O-[3-O-o-anisoyl]- α-l-rhamnopyranoside (**4**)

Yellow powder. ${\left[\alpha \right]}_{D}^{23}$ = −161.64° (c = 0.28, MeOH). HR ESI-MS: *m*/*z* 1021.3170 [M + H]^+^ (calcd. for [C_47_H_57_O_25_]^+^, 1021.3189). UV λmax/nm (log ε): 268 (5.15), 345 (5.01). IR νmax/cm^−1^: 1601, 3430. 1H and 13C NMR spectra were recorded in CD_3_OD on Bruker AV600 (600 MHz) instruments (Bruker, Massachusetts, USA): see Table 1.

### 3.9. Quantitative Analysis

F50 and F80 samples were dissolved in methanol to a concentration of 10 mg/mL, and the solutions were injected (20 μL) into a reverse-phase HPLC-PDA system described above. The mobile phase comprised 0.5% HCOOH in H_2_O (A) and 0.5% HCOOH in acetonitrile (B), and the following gradient was used: initial 10% B for 10 min, 15% B at 30 min, 25% B at 45 min, 35% B at 55 min, 45% B at 60 min, and 55% B at 70 min. The flow rate was set to 1 mL/min, and detection was performed at 330 nm corresponds to the absorbance of the flavonol backbone. Quantification was performed using the area under each peak determined by ChromNAV software (JASCO, Tokyo, Japan). Kaempferol compounds diluted in methanol with concentrations of 0.0017, 0.0035, 0.0175, 0.0349, 0.1747, 0.3494, and 3.4937 μM were used as standards. The content of each compound was expressed as milligram per 100 g dry flour weight.

### 3.10. In Vitro α-Glucosidase Inhibition Activity

The α-glucosidase inhibitory activity was assayed in vitro using a 96-well plate as reported earlier [30] with Versa max microplate reader (Molecular Devices, San Jose, CA, USA). Briefly, 1 mg of each sample and acarbose were dissolved in 20% of DMSO in phosphate buffer (50 mM, pH 6.9). The 50 μL/well of sample solution mixed with 20 μL/well of α-glucosidase (0.5 U/mL; diluted in phosphate buffer) from *Saccharomyces*
*cereviceae* (Type I, Sigma-Aldrich, St. Louis, MO, USA). After preincubated at 37 °C for 15 min, the assay was initiated by adding 50 μL/well of 4-nitrophenyl-α-D-glucopyranoside (PNPG) (9 mg/mL; diluted in phosphate buffer) as a substrate. Then, the mixture was incubated further at 37 °C for 15 min. The reaction was stopped by adding 100 μL/well of Na_2_CO_3_ (100 mM; diluted in phosphate buffer). The absorbance of the released *p*-nitrophenol was measured at 415 nm. Solution of 20% DMSO in phosphate buffer was set up as a control and each experiment was performed in triplicate. The results were expressed as percent inhibition and calculated using the formula: inhibitory activity (%) = (1 − (Abs sample/Abs control) × 100%

Statistical analysis was performed using the analysis of variance (ANOVA) and the Tukey’s HSD post hoc test carried out at significance level of *p* < 0.05 to determine differences between the samples using Past 4.0 software (Oslo, Norway) [31]. 

## 4. Conclusions

To enhance the added value and expand the application scope of jack bean, we focused on the isolation and identification of polyphenols contained in this legume. As a result, four kaempferol glycosides were isolated, one of which was novel, while the others have not been previously found in jack bean. Furthermore, ten more compounds (all of them kaempferol glycosides) were identified by LC-MS/MS. These compounds were exhibited by the inhibition activity of α-glucosidase. The presence of anisoyl group in the kaempferol glycosides made these compounds potential for preventing diabetes. These results suggested that jack bean can be utilized as a raw ingredient for foods and a possible source of functional compounds. These findings highlight the importance of daily flavonoid intake with food and demonstrating that jack bean is a potential source of flavonoids, especially kaempferol. Furthermore, these suggestions also underscore the necessity of investigating the effects of the food processing applied. Considering these bioactive compounds were usually unstable under heating condition, the proper application of this bean as a food ingredient need to be found for the optimum efforts to reduce levels of antinutritional compounds and minimize degradation of these bioactive compounds.

## Figures and Tables

**Figure 1 molecules-25-02481-f001:**
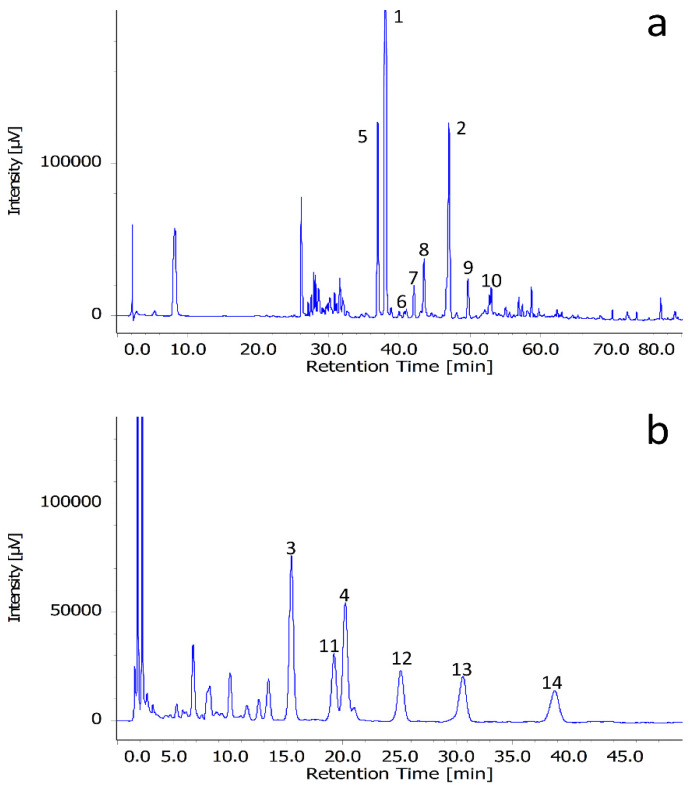
Chromatograms of F50 (**a**) and F80 (**b**).

**Figure 2 molecules-25-02481-f002:**
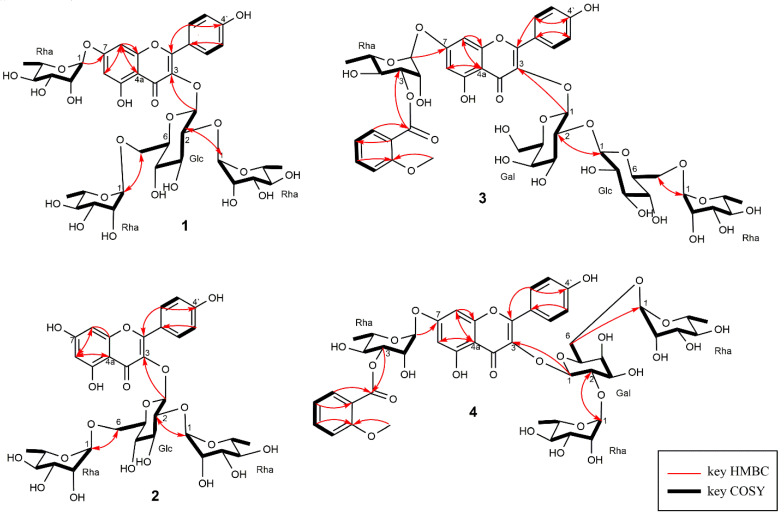
Chemical structures of compounds **1**–**4** isolated from *Canavalia ensiformis*. HMBC (heteronuclear multiple bond correlation spectroscopy), COSY (homonuclear correlation spectroscopy).

**Figure 3 molecules-25-02481-f003:**
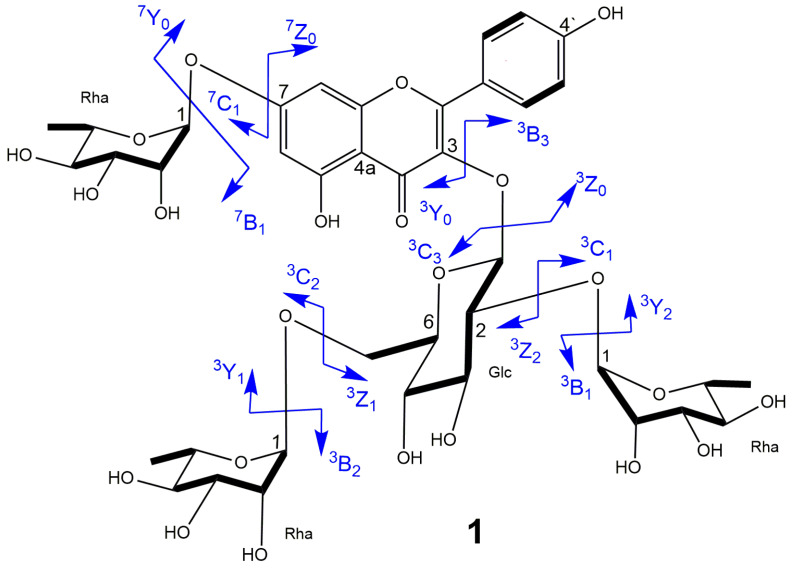
Fragmentation patterns of compound **1**.

**Table 1 molecules-25-02481-t001:** ^1^H and ^13^C NMR data of compounds **1**–**4** recorded in CD_3_OD.

C	1 ^a^		2 ^a^		3 ^a^		4 ^b^	
	δ_H_ (mult., *J* = Hz)	δ_C_	δ_H_ (mult., *J* = Hz)	d_C_	δ_H_ (mult., *J* = Hz)	δ_C_	δ_H_ (mult., *J* = Hz)	δ_C_
**Aglycone**								
2	-	159.22	-	158.64	-	161.79	-	159.22
3	-	134.72	-	134.40	-	135.15	-	134.73
4	-	179.52	-	179.40	-	179.94	-	179.50
5	-	157.92	-	163.13	-	162.87	-	163.17
6	6.36 (d, 2.4)	100.42	6.19 (d, 2.0)	99.75	6.55 (s)	100.70	6.52 (s)	100.30
7	-	163.40	-	165.56	-	163.40	-	163.17
8	6.63 (d,2.4)	95.62	6.38 (d, 2.0)	94.64	6.82 (s)	95.91	6.78 (s)	95.66
9	-	162.90	-	158.40	-	158.00	-	157.95
10	-	107.50	-	105.90	-	107.30	-	107.68
1′		122.80		123.00		122.40		122.80
2′, 6′	8.00 (d, 8.8)	132.33	8.06 (d, 8.8)	132.30	8.13 (d, 8.8)	132.65	8.11 (d, 8.6)	132.34
3′, 5′	6.81 (d, 8.8)	116.25	6.90 (d, 8.8)	116.20	6.93 (d, 8.8)	116.33	6.92 (d, 8.2)	116.23
4′	-	161.50	-	161.30	-	159.64	-	161.40
**3-O-Glc/3-O-Gal^c^**								
1	5.53 (d, 8.0)	100.82	5.60 (d, 8.0)	100.83	5.28 (d, 7.0)	101.64	5.63 (br s)	100.79
2	3.84 (br s)	77.57	3.92 (dd, 9.6, 8.0)	77.54	4.07 (dd, 7.8, 9.4)	80.14	3.95 (t, 8.4)	77.58
3	3.61 (dd, 3.2, 9.6)	75.69	3.69 (dd, 3.2, 9.6)	75.70	3.73 (m)	74.74	3.70 (br s)	75.58
4	3.68 (d, 3.2)	70.67	3.76 (d, 3.2)	70.68	3.81 (m)	70.92	3.78 (d, 3.0)	70.63
5	3.54 (t, 6.4)	75.34	3.63 (t, 6.4)	75.29	3.35 (br s)	78.20	3.48 (br s)	72.34
6	3.36 (m); 3.68 (m)	67.15	3.49 (m); 3.72 (m)	67.11	3.70 (m)	62.56	3.65 (m), 3.70 (m)	67.14
**2′′-O-Rha/2′′-O-Glc^d^**								
1	5.12 (s)	102.59	5.21 (br s)	102.60	4.77 (d, 7.2)	104.80	5.25 (s)	102.54
2	3.91 (br s)	71.68	4.00 (m)	72.40	3.41 (m)	71.23	4.01 (br s)	72.34
3	3.69 (dd, 3.2, 9.6)	72.37	3.80 (dd, 3.2, 9.6)	72.44	3.42 (m)	77.85	3.82 (br s)	72.01
4	3,25 (t, 9.6)	74.02	3.34 (t, 5.6)	74.06	3.40 (d, 2.4)	75.56	3.40 (m)	74.03
5	3.95 (dd, 6.4, 9.6)	69.83	4.06 (m)	69.81	3.61 (t, 6.0)	75.26	3.50 (m)	69.79
6	0.88 (d, 6.4)	17.54	0.97 (d, 5.6)	17.53	3.65 (br s); 3.40 (m)	67.21	1.17 (d, 6.0)	17.94
**6′′-O-Rha/1′′′-O-Rha**								
1	4.42 (s)	101.80	4.52 (s)	101.86	4.51 (br s)	101.83	4.51 (br s)	101.78
2	3.45 (br s)	71.68	3.56 (br s)	72.07	3.55 (br s)	72.03	3.55 (br s)	72.25
3	3.40 (m)	72.23	3.51 (m)	72.27	3.48 (dd, 9.6, 4.0)	72.26	3.46 (d, 5.4)	67.14
4	3.21 (m)	73.86	3.27 (m)	73.85	3.27 (t, 9.6)	73.86	3.29 (br s)	73.86
5	3.41 (m)	69.66	3.53 (m)	69.68	3.51 (m)	69.67	4.06 (m)	69.63
6	1.07 (d, 6.4)	17.95	1.17 (d, 5.6)	17.94	1.17 (d, 5.6)	17.94	0.99 (d, 6.0)	17.54
**7-O-Rha**								
1	5.47 (s)	99.87	-	-	5.63 (s)	99.82	5.62 (s)	99.81
2	3.93 (br s)	71.68	-	-	4.35 (br s)	69.50	4.35 (br s)	69.52
3	3.73 (dd, 3.2, 9.6)	72.04	-	-	5.34 (dd, 9.6, 3.2)	75.42	5.35 (dd, 9.0, 3.6)	75.62
4	3.39 (br s)	73.59	-	-	3.84 (t, 9.6)	70.92	3.81 (br s)	70.91
5	3.51 (dd, 5.6, 9.6)	71.24	-	-	3.79 (m)	71.51	3.80 (m)	71.45
6	1.16 (d, 5.6)	18.10	-	-	1.33 (d. 6.4)	18.15	1.33 (d, 6.0)	18.14
**Anis**								
-1’	-	-	-	-	-	120.96	-	121.14
-2’	-	-	-	-	-	160.81	-	160.76
-3’	-	-	-	-	7.13 (d, 8.8)	113.37	7.13 (d, 8.4)	113.37
-4’	-	-	-	-	7.55 (t, 7.2)	136.15	7.55 (t, 7.8)	135.19
-5’	-	-	-	-	7.03 (t, 8.0)	121.15	7.03 (t, 7.8)	121.01
-6’	-	-	-	-	8.00 (d, 6.4)	132.91	7.96 (d, 7.8)	131.48
-OCH_3_	-	-	-	-	3.90 (s)	56.43	3.90 (s)	56.45
-CO	-	-	-	-	-	167.38	-	167.37

^a^ Determined by 800-MHz NMR spectroscopy, ^b^ Determined by 600-MHz NMR spectroscopy, ^c^ Glc in **1** and **2**; Gal in **3** and **4**, ^d^ Rha in **1**, **2**, and **4**; Glc in **3**.

**Table 2 molecules-25-02481-t002:** MS/MS fragment ions of compounds **1**–**4**.

Compound	Fragment Ions ^a^								
	^3^Y_2_	^3^Y_1_	^3^Y_0_	^3^Z_2_	^3^Z_1_	^3^Y_0_^7^Y_2_	^3^Y_0_^7^Y_1_	^3^Y_0_^7^Y_0_	[A+R]^+ b^
1	741.2214	595.1542	433.0958	n.d ^c^	579.1572	n.d	n.d	287.0376	−
2	595.1542	449.1033	287.0376	n.d	433.0958	−	−	−	−
3	891.2531	729.1957	567.1334	875.2322	n.d	n.d	433.1136	287.0576	281.1014
4	875.2546	729.2129	567.1486	n.d	713.1918	433.1136	n.d	287.0576	281.1014

^a^ positive mode, ^b^ A = anis; R = rhamnose, ^c^ n.d = not detected.

**Table 3 molecules-25-02481-t003:** Characterization of compounds **1–14**.

Compound	Content (mg/100 g dry wt.) ^a^	Formula	[M + H]+			Aglycone ^b^	Derivatives ^c^		Detection
			Calculated Mass	Exact Mass	Mass Error(ppm)		Glycans	Other	
1	2.2862 ± 0.0745	C_39_H_50_O_23_	887.2821	887.2843	2.4795	K	Rha, Rha, Rha, Hex		MS, NMR
2	0.6651 ± 0.0205	C_33_H_40_O_19_	741.2242	741.2214	−3.7775	K	Rha, Rha, Hex		MS, NMR
3	0.4906 ± 0.0176	C_47_H_56_O_26_	1037.3138	1037.3076	−5.9770	K	Rha, Rha, Hex, Hex	Anis	MS, NMR
4	0.3916 ± 0.0134	C_47_H_56_O_25_	1021.3189	1021.3170	−1.8603	K	Rha, Rha, Rha, Hex	Anis	MS, NMR
5	0.5743 ± 0.0185	C_39_H_50_O_24_	903.2770	903.2747	−2.5463	K	Rha, Rha, Hex, Hex		MS
6	0.0778 ± 0.0033	C_38_H_48_O_23_	873.2665	873.2609	−6.4127	K	Rha, Rha, Hex, Xyl		MS
7	0.1987 ± 0.0072	C_33_H_40_O_20_	757.2191	757.2161	−3.9617	K	Rha, Hex, Hex		MS
8	0.0139 ± 0.0031	C_33_H_40_O_19_	741.2242	741.2214	−3.7775	K	Rha, Rha, Hex		MS
9	0.1089 ± 0.0041	C_32_H_38_O_19_	727.2086	727.2133	6.4631	K	Rha, Hex, Xyl		MS
10	0.0520 ± 0.0015	C_27_H_30_O_15_	595.1663	595.1622	−6.8888	K	Rha, Hex		MS
11	0.3045 ± 0.0162	C_47_H_56_O_26_	1037.3138	1037.3076	−5.9770	K	Rha, Rha, Hex, Hex	Anis	MS
12	0.2475 ± 0.0105	C_47_H_56_O_25_	1021.3189	1021.3170	−1.8603	K	Rha, Rha, Rha, Hex	Anis	MS
13	0.0866 ± 0.0037	C_47_H_56_O_26_	1037.3138	1037.3076	−5.9770	K	Rha, Rha, Hex, Hex,	Anis	MS
14	0.0722 ± 0.0030	C_47_H_56_O_25_	1021.3189	1021.3170	−1.8603	K	Rha, Rha, Rha, Hex	Anis	MS

^a^ Values were expressed as mean ± SD (n = 4), ^b^ K = kaempferol, ^c^ Rha = rhamnosyl; Gal = galactosyl; Glu = glucosyl; Hex = hexose, Xyl = xylosyl; Anis = anisoyl.

**Table 4 molecules-25-02481-t004:** In vitro α-glucosidase inhibition activity of compounds **1**–**4**.

Sample *	Inhibition Activity (%) **
F20	trace ***
F50	24.91 ± 0.84^a^
F80	51.34 ± 1.23^d^
F100	trace
Compound **1**	30.96 ± 1.62^b^
Compound **2**	39.54 ± 0.69^c^
Compound **3**	90.07 ± 1.60^g^
Compound **4**	77.06 ± 0.79^f^
Acarbose	71.71 ± 0.08^e^

* purity of **1**–**4** were depicted at Appendix A, ** value expressed as mean ± SD (n = 3). value followed with different letter showed statistical significant different at *p* < 0.05 (Tukey’s test), *** inhibition activity was not detected in the analysis sample concentration (1 mg/mL).

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
