# Peer review of "Identification and Characterization of α-Glucosidase Inhibition Flavonol Glycosides from Jack Bean (Canavalia ensiformis (L.) DC"

_molecules, 2020, doi:10.3390/molecules25112481_

Round 1

Reviewer 1 Report

This manuscript reports identification of kaempherol derivatives in Jack vean extracts. Detailed description of data obtained with a survey of appropriate techniques is afforded. Discussion of data on compound identification is well conducted and reasoned.

There is only an issue that should be addressed. Authors should discuss in more detailed the concerns relative to the Alpha-glucosidase assay for their suggestion of anti-diabetic use of the plant extracts as proposed in the title, or otherwise they should change the tiltle to relate the results on Alpha-glucosidase inhibition rather than anti-diabetic functionality. Indeed, according to the results presented by the authors on this issue, only compounds 3 and 4 have a high enzyme inhibition, which rises concerns on such capacity for the all plant extract.

I have seen a note on the section 3. Discussion which I do not know as to the note comes from a previous reviewer or the editors.

Author Response

We thank Referees for careful reading our manuscript and for giving useful comments. We have revised the manuscript on the basis of the Referee’s comments.

  1. Authors should discuss in more detailed the concerns relative to the Alpha-glucosidase assay for their suggestion of anti-diabetic use of the plant extracts as proposed in the title, or otherwise they should change the title to relate the results on Alpha-glucosidase inhibition rather than anti-diabetic functionality. Indeed, according to the results presented by the authors on this issue, only compounds 3 and 4 have a high enzyme inhibition, which rises concerns on such capacity for the all plant extract.

Thank you for useful comment for our manuscript.

We have changed the title of the article to “Identification and characterization of a-glucosidase inhibition flavonol glycosides from jack bean (Canavalia ensiformis (L.) DC.” so that it has more related with the results of the activity of a-glucosidase. And also, the explanation about the α-glucosidase and diabetic disease were added in introduction section (L42-47).

As according to the explanation in sub section 2.4, the inhibitory activity of a-glucosidase for compounds 3 and 4 were higher than acarbose and is thought to be influenced by the presence of anisoyl groups. Thus, we estimate that another kaempferol glycoside compounds which have an anisoyl group in its structure (compounds 11 14) also has a high a-glucosidase inhibitory activity. This group of compounds constitutes 28.60% of the total kaempferol glycoside contained by the jack bean. We had added this discussion in the section 2.4 (L236-246).

  1. I have seen a note on the section 3. Discussion which I do not know as to the note comes from a previous reviewer or the editors.

We regret this oversight. The reviewer's comment is correct. We had deleted this note.

Thank you again for your comments on our paper. We look forward to a publication of our manuscript in Molecules

Reviewer 2 Report

The manuscript entitled “Identification and characterization of antidiabetic flavonol glycosides from jack bean (Canavalia ensiformis (L.) DC.”, even if not presenting a great impact, is enough interesting keeping a sufficient level of novelty and originality. The work follows a very traditional scheme of “separation and characterization” applied in chemistry of natural substances, with the use of powerful spectroscopical devices that ease the task. All experimental parts have been performed with a good methodological approach, and results sound coherent.

I suggest few minor considerations/corrections:

-authors correctly report that the use of jack beans is strongly limited by partial toxic and antinutritional composition, but that the content of these ingredients can be easily reduced by heating (in truth some authors report just a partial toxicity lowering). On the other hand, the claimed potential use of jack beans as antidiabetic resides (according to the proposed biological tests) in the presence of glycosylated polyphenols. Do the authors imagine that in the reported necessary heating conditions (to “lower” toxicity), these compounds will be really stable and no process, such as oxidation, hydrolysis or even polymerization, will occur? Authors can well understand that If it should happen the claim of antidiabetic activity, as here reported, would no more be real.

-a general check of typos and editorial format is necessary; since authors report together the “results and discussion” part, paragraph 3 has to be deleted and all the following renumbered in accordance.

Author Response

To the Reviewer 2:

We thank Referees for careful reading our manuscript and for giving useful comments. We have revised the manuscript on the basis of the Referee’s comments.

  1. Authors correctly report that the use of jack beans is strongly limited by partial toxic and antinutritional composition, but that the content of these ingredients can be easily reduced by heating (in truth some authors report just a partial toxicity lowering). On the other hand, the claimed potential use of jack beans as antidiabetic resides (according to the proposed biological tests) in the presence of glycosylated polyphenols. Do the authors imagine that in the reported necessary heating conditions (to “lower” toxicity), these compounds will be really stable and no process, such as oxidation, hydrolysis or even polymerization, will occur? Authors can well understand that If it should happen the claim of antidiabetic activity, as here reported, would no more be real.

We appreciate the reviewer's concerns on this point. It is generally known that some of the polyphenol is unstable under heating condition. We are interested in the chemically changes of the glycosylated polyphenols in this bean during food processing, such as necessary heating conditions to “lower” toxicity. This research is in progress, we would like to report on this point as the next our research. On the other hand, it has been attempted to use non heat treatment such as germination and using some chemicals. We think these processing methods might be useful for utilization of this bean. We added these point in introduction (L63-65) and conclusion section. 

  1. a general check of typos and editorial format is necessary

The reviewer's comment is accurate. We corrected according to this comment (i.e L182, 192-193, 195, 197, 204, 209).

  1. since authors report together the “results and discussion” part, paragraph 3 has to be deleted and all the following renumbered in accordance

Thank you for your kind suggestion. We have removed paragraph 3 and rewritten as “results and discussion” section.

Thank you again for your comments on our paper. We look forward to a publication of our manuscript in Molecules.

Reviewer 3 Report

The manuscript shows an analytical study of polyphenols in defatted flour of jack beans. By different analytical methods (UV-visible, 1D, 2D NMR, and HR-ESI-MS) the authors identified 14 polyphenols, all glycosides of kaempferol, and their chemical structures were characterized. The four main polyphenols were quantified and tested for inhibitory enzymatic activity on alpha-glucosidase.

The study has novelty degree, the experimental design is well performed and the results are clearly showed. However, some improvements can be made into the discussion of the results.

I have some suggestions to improve the quality of the manuscript as follows:

Please explain all abbreviations used in the manuscript when these appear for the first time in the text.

Please add a sub-section for Reagents and one for Statistical analysis in the section 2.

Line 156          ”Figures” instead of ”Figure”

Line 165          ” sufficient quantities” instead of ”quantities sufficient”

Line 166          compounds 1-4

Line 171          Figures S25-S34

Lines 179-182  There are too many digits after the decimal point; two should be enough.

Line 194          compounds 1-4

Line 195          ”statistical significant diference” instead of ”significantly different”. Please add in brackets the method used for statistical analysis.

Lines 207-210  Please delete these lines! They seem to be comments of a reviewer. Indeed, I’m in agreement with these suggestions.

Lines 228-229  Please explaine more detailed how did you obtain the five sub-fractions and how did you analyze them.

Line 230          Why did you use TLC? Only for the sub-fractions of F50?

Line 248          ”Chromatograms of F50 and F80 are shown in Figure 1a and 1b.” instead of ” Chromatogram of F50 and F80 as shown on Figure 1a and 1b.”

Line 315          ”Quantification” instead of ”Quantitation”

Line 317          What solvent did you use for standard solutions?

Line 322          And acarbose? The high concentration of DMSO (20%) didn’t affect the enzyme activity?

Lines 325 and 327       What solvent did you use for PNPG and Na2CO3?

Line 339          Please rephrase the sentence starting with „These findings... „. The sentence is too long.

Author Response

To the Reviewer 3:

We thank Referees for careful reading our manuscript and for giving useful comments. We have revised the manuscript on the basis of the Referee’s comments.

  1. Please explain all abbreviations used in the manuscript when these appear for the first time in the text.

We regret this oversight. The reviewer's comment is correct. We have revised as suggested by the reviewer (i.e L80-82, 90-91, 93).

  1. Please add a sub-section for Reagents and one for Statistical analysis in the section 2.

We agree that this point requires clarification.  We added sub-section 3.2 for reagents and standards used in this research (L255-271). Statistical analysis was used for a-glucosidase inhibition activity assay, therefore we added the statistical analysis information in section 3.10 (L389-391).

  1. Line 156: ”Figures” instead of ”Figure”

This error has been corrected in accordance with the reviewer's comment. (L182).

  1. Line 165: ”sufficient quantities” instead of ”quantities sufficient”

This error has been corrected in accordance with the reviewer's comment (L191).

  1. Line 166: compounds 1-4

This error has been corrected in accordance with the reviewer's comment (L192-193).

  1. Line 171: Figures S25-S34

This error has been corrected in accordance with the reviewer's comment (L197).

  1. Lines 179-182: There are too many digits after the decimal point; two should be enough.

This error has been corrected in accordance with the reviewer's comment (L204-205).

  1. Line 194: compounds 1-4

This error has been corrected in accordance with the reviewer's comment (L230).

  1. Line 195: ”statistical significant difference” instead of ”significantly different”. Please add in brackets the method used for statistical analysis

We have revised as suggested by the reviewer and added the Tukey’s test as statistical analysis (L225-226).

  1. Lines 207-210:  Please delete these lines! They seem to be comments of a reviewer. Indeed, I’m in agreement with these suggestions.

We regret this oversight. The reviewer's comment is correct. We had deleted this note.

  1. Lines 228-229:  Please explained more detailed how did you obtain the five sub-fractions and how did you analyze them.

We agree that explanation of isolation method was not clear, have revised this section. Sub-fraction F50A–F50E were obtained from fraction F50 using silica gel open-column chromatography.  Sub-fractions F50A and F50B were pure compounds which are compound 1 and 2 each. Detail were written on L282-286. TLC was performed to monitor the collection of fractions (L284-286). Isolation for compound 3 and 4 were performed using preparative HPLC. Detail were written on L287-294.

  1. Line 230: Why did you use TLC? Only for the sub-fractions of F50?

TLC was performed to monitor the collection of fractions (L284-285). We agree that explanation of isolation method was not clear, have revised this section (L282-286).

  1. Line 248: ”Chromatograms of F50 and F80 are shown in Figure 1a and 1b.” instead of ” Chromatogram of F50 and F80 as shown on Figure 1a and 1b.”

We have revised as suggested by the reviewer (L304).

  1. Line 315: ”Quantification” instead of ”Quantitation”

We have revised as suggested by the reviewer (L371).

  1. Line 317: What solvent did you use for standard solutions?

We use methanol for making the standard solution. We added this information at L373.

  1. Line 322: And acarbose? The high concentration of DMSO (20%) didn’t affect the enzyme activity?

We have revised as suggested by the reviewer. Sample and acarbose were dissolved in 20% DMSO in phosphate buffer (50 mM, pH 6.9)( (L379). To consider the effect of DMSO to the enzyme activity, 20%DMSO in phosphate buffer solution was also added to the control and was evaluated again. (L385-387).

  1. Lines 325 and 327: What solvent did you use for PNPG and Na2CO3?

We use phosphate buffer (50 mM, pH 6.9) for PNPG (L383) and Na2CO3 (L384-385). We added this information.

  1. Line 339:  Please rephrase the sentence starting with „These findings... „. The sentence is too long

We have revised the sentence and added several sentences to improve the content of conclusion section (L400-406).

Thank you again for your comments on our paper. We look forward to a publication of our manuscript in Molecules.

Reviewer 4 Report

In this manuscript, the authors investigated the flavonol glycoside profile of jack bean and identified major jack bean flavonol components using NMR and/or LC-MS/MS. Four main components which were characterized as kaempferol glycosides were isolated and purified, and their anti-diabetic activities were evaluated through in vitro a-glucosidase inhibition assay.

Main critiques:

Introduction –

Introduction is too brief. More background information on the related researches of jack bean flavonols should be included.

Materials and Methods –

Isolation and purification details on compounds 1-4 should be added. TLC was used to isolate compounds 1 and 2 while prep-HPLC was used for compounds 3 and 4. Details on their isolation is missing.

Results –

Identification details on compounds other than 1-4 should be added (MS fragmentation details and their interpretation). Accordingly, in table 2, more MS data should be provided, such as the fragmentation patterns and mass errors.

Kaempferol aglycone was used to quantify its glycosides identified in Table 2. The quantification data was compared with other studies on other materials (section 2.3). If the quantification methods are not consistent (i.e. different standards were used), justification on such comparison is needed.

Purity of isolated compounds 1-4 analyzed in Table 3 was not provided.  

Table 3, consider adding more analysis on compound fractions (i.e. F50 and F80 in Figure 1) and comparing the activities between fractions and individual compounds 1-4.

Discussion –

No real discussion was provided.  

Author Response

To the Reviewer 4:

We thank Referees for careful reading our manuscript and for giving useful comments. We have revised the manuscript on the basis of the Referee’s comments.

  1. Introduction: Introduction is too brief. More background information on the related researches of jack bean flavonols should be included.

Thank you for useful comment for our manuscript.

We have revised the introduction section as suggested by the reviewer (L34-55).

  1. Materials and Methods: Isolation and purification details on compounds 1-4 should be added. TLC was used to isolate compounds 1 and 2 while prep-HPLC was used for compounds 3 and 4. Details on their isolation is missing.

Thank you for useful comment for our manuscript.

We have revised as suggested by the reviewer. We agree that explanation of isolation method was not clear, have revised this section.

Isolation for compounds 1 and 2 were performed using silica gel open-column chromatography. Detail were written on L282-286. TLC was performed to monitor the collection of fractions (L284-285).

Isolation for compound 3 and 4 were performed using preparative HPLC. Detail were written on L287-294.

  1. Results: Identification details on compounds other than 1-4 should be added (MS fragmentation details and their interpretation). Accordingly, in table 2, more MS data should be provided, such as the fragmentation patterns and mass errors.

We have revised as suggested by the reviewer. We added information about MS/MS fragment ions of compounds 14 were written in Table 2.   Information about calculated mass, exact mass and error were added in Table 3.

  1. Kaempferol aglycone was used to quantify its glycosides identified in Table 2. The quantification data was compared with other studies on other materials (section 2.3). If the quantification methods are not consistent (i.e. different standards were used), justification on such comparison is needed.

Thank you for useful comment for our manuscript. We agree with the reviewer. Because the quantification methods were different each report, discussion about the quantitate comparison with other materials were deleted. New discussion about comparison of each kaempferol derivatives in the jack bean was added (L202-208).

  1. Purity of isolated compounds 1-4 analyzed in Table 3 was not provided.

We added chromatogram of compounds 14 to describe the purity of each compound in Figures S42-S43 as the supplementary data.

  1. Table 3, consider adding more analysis on compound fractions (i.e. F50 and F80 in Figure 1) and comparing the activities between fractions and individual compounds 1-4.

Thank you for useful comment for our manuscript. We reanalyzed the inhibition activity of a-glucosidase for 4 fractions (F20, F50, F80 and F100), compounds 14 and acarbose.  The result was written in Table 4 (L224).

  1. Discussion: No real discussion was provided.

  In accordance with other reviewer's comment, we have removed “discussion” section and rewritten as “results and discussion” section. And discussion about quantity of kaempferol glycoside and glucosidase inhibition activity were added in each section.

Thank you again for your comments on our paper. We look forward to a publication of our manuscript in Molecules.

Round 2

Reviewer 4 Report

In this revised manuscript the authors have properly addressed all the previous comments. I recommend the manuscript to be published in Molecules.